# Reconstructing disease dynamics for mechanistic insights and clinical benefit

Amit Frishberg[1,2,3,4,11], Neta Milman[1,11], Ayelet Alpert[1], Hannah Spitzer[2,5], Ben Asani[6], Johannes B. Schiefelbein[6], Evgeny Bakin[4], Karen Regev-Berman[4], Siegfried G. Priglinger[6], Joachim L. Schultze[3,7,8], Fabian J. Theis[2,9,10,12] & Shai S. Shen-Orr[1,4,12] ✉

Diseases change over time, both phenotypically and in their underlying molecular processes. Though understanding disease progression dynamics is critical for diagnostics and treatment, capturing these dynamics is difficult due to their complexity and the high heterogeneity in disease development between individuals. We present TimeAx, an algorithm which builds a comparative framework for capturing disease dynamics using high-dimensional, short time-series data. We demonstrate the utility of TimeAx by studying disease progression dynamics for multiple diseases and data types. Notably, for urothelial bladder cancer tumorigenesis, we identify a stromal pro-invasion point on the disease progression axis, characterized by massive immune cell infiltration to the tumor microenvironment and increased mortality. Moreover, the continuous TimeAx model differentiates between early and late tumors within the same tumor subtype, uncovering molecular transitions and potential targetable pathways. Overall, we present a powerful approach for studying disease progression dynamics—providing improved molecular interpretability and clinical benefits for patient stratification and outcome prediction.

Diseases are dynamic processes encompassing a multitude of changes. These range from intra-cellular molecular states, such as those occurring following cellular differentiation or activation, to changes in systemic molecular, cellular and physiological states. Identifying the underlying dynamics of diseases at high-resolution enables their quantitative comparison and is critical for designing novel preventive and therapeutic strategies to improve health. Time-series experimental designs provide an opportunity for studying disease dynamics and the variability across patients. However, while disease progression rate is patient-specific, time-series data is usually collected at fixed intervals. This reduces the efficiency of comparing progression dynamics when using time as a predictive variable and forces the clustering of data from different time points to obtain some level of shared dynamics (Fig. 1A). Such attempts are further confounded by the fact that disease progression dynamics are often orchestrated by multiple biological processes simultaneously, requiring modeling to

[1]Department of Immunology, Faculty of Medicine, Technion-Israel Institute of Technology, Haifa, Israel. [2]Institute of Computational Biology, Helmholtz Center Munich, 85764 Neuherberg, Germany. [3]Systems Medicine, Deutsches Zentrum für Neurodegenerative Erkrankungen (DZNE), Bonn, Germany. [4]CytoReason, Tel-Aviv, Israel. [5]Institute for Stroke and Dementia Research (ISD), LMU University Hospital, LMU Munich, Germany. [6]Department of Ophthalmology, Ludwig-Maximilians-University, Munich, Germany. [7]Genomics and Immunoregulation, Life & Medical Sciences (LIMES) Institute, University of Bonn, Bonn, Germany. [8]Deutsches Zentrum für Neurodegenerative Erkrankungen (DZNE). PRECISE Platform for Genomics and Epigenomics at DZNE and University of Bonn, Bonn, Germany. [9]Department of Mathematics, Technical University of Munich, 85748 Garching, Germany. [10]Technical University of Munich, TUM School of Life Sciences Weihenstephan, 85354 Freising, Germany. [11]These authors contributed equally: Amit Frishberg, Neta Milman. [12]These authors jointly supervised this work: Fabian J. Theis, Shai S. Shen-Orr. ✉e-mail: shenorr@technion.ac.il

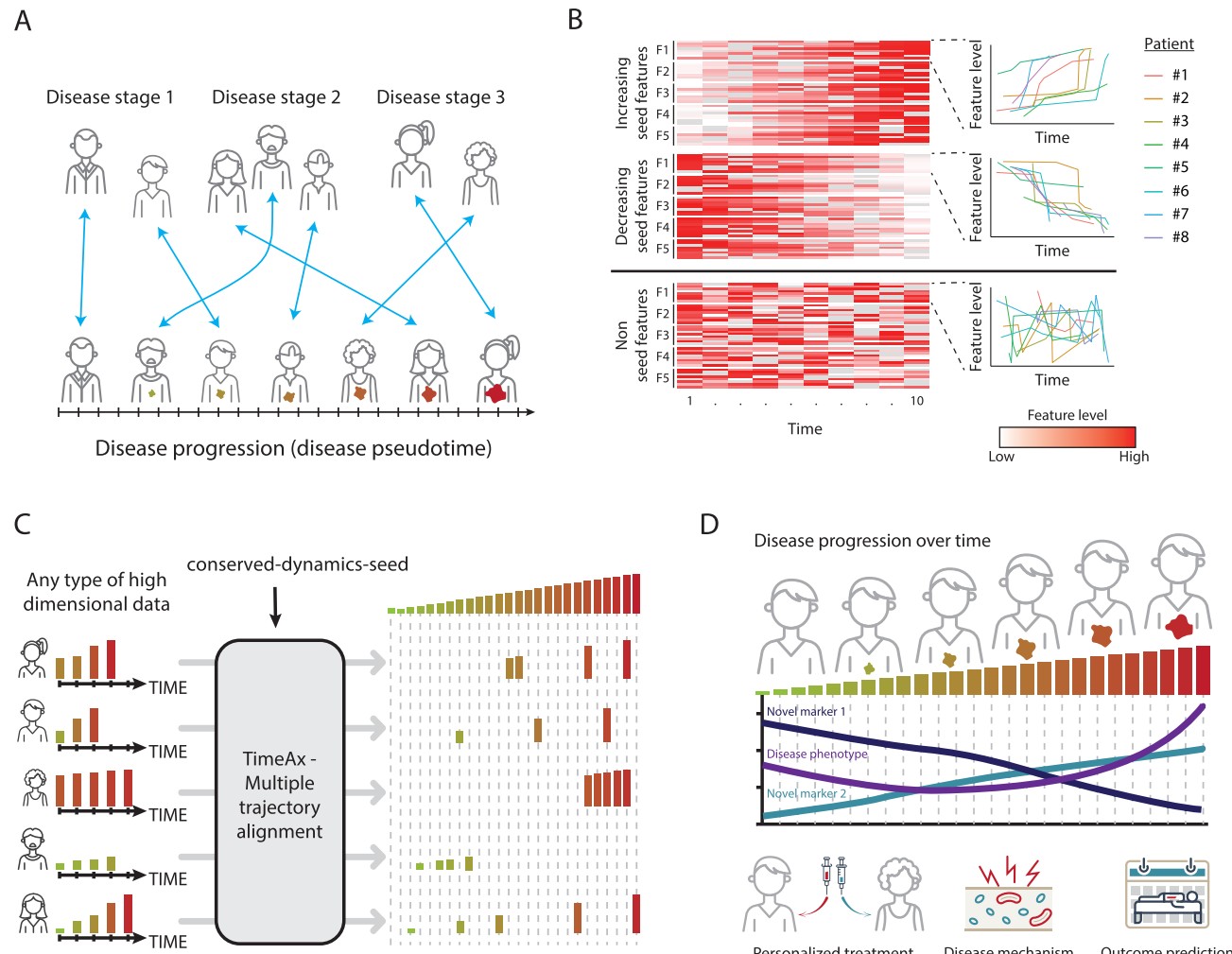

**Fig. 1 | TimeAx discovers the shared disease dynamics across multiple patients.**
**A** Disease progression is not captured by patient subtyping. While current patient stratification and disease state comparison requires clustering of patients into coarse subgroups (top), understanding disease dynamics as a common ground between patients' disease trajectories enables patient stratification in a higher resolution (bottom). **B** TimeAx seed features selection process. TimeAx selects a *"conserved-dynamics-seed"* with shared dynamics across all subjects. In the heatmap (left), each row is a patient and each column is a sample collected at a different time. Each feature spans multiple patients (multiple rows) and is also presented as a line plot (right). **C** An illustration describing TimeAx's utility for revealing the shared disease progression dynamics and the projection of patients-specific disease pseudotime positions, integrating the differences between patients' disease trajectories over time. **D** Disease pseudotime can be utilized to discover novel disease mechanisms as well as to support new clinical frameworks for patient stratification and outcome prediction. We thank Yuval Abraham for his contribution in the design and creation of (**A**, **C** and **D**).

be performed in a high dimensional space. Dimensionality reduction methods, such as Principal Component Analysis (PCA), Factor Analysis (FA)[1], as well as state-of-the-art trajectory inference methods such as diffusion maps (DMs)[2], may be driven by time-independent variation, yielding low interpretability and making the inference of patients' disease progression dynamics non trivial[3]. Recent advances in this area focused on the calculation of differentially expressed features over time, but only when comparing extreme states, such as disease versus controls[4,5], rather than describing the global dynamics of the disease. Therefore, forming a model for disease progression remains challenging, even when large time-series data are available.

Here, we present TimeAx, a method that captures a representation of disease dynamics over time based on time-series data from multiple individuals. Akin to multiple sequence alignment, a historically transformative tool, which enabled biologists to build a quantitative-mechanistic understanding of DNA and protein functions[6], TimeAx performs multiple trajectory alignment providing major benefits for molecular interpretation and clinical diagnosis of both acute and chronic diseases. We demonstrate the utility of TimeAx for multiple time-series data types, including patients' transcriptomes

and features extracted from medical imaging. Overall, our framework allows for a high resolution understanding of disease progression dynamics, discovery of underlying molecular mechanisms, and paves the way for better clinical decision making—including the design of new clinical interventions.

## Results

### TimeAx reveals shared disease dynamics across multiple patients

Patient cohorts tend to display high heterogeneity in patients' disease courses, masking shared disease progression dynamics and the underlying biological mechanisms that are shared across all patients. Often, the naive solution is clustering patients into disease-specific, clinical stages or subtypes, which many times fails to capture the continuous dynamics of the disease and its progression over time (Fig. 1A). To overcome this problem, TimeAx quantitatively models a representation of the shared disease dynamics over time. TimeAx relies solely on measured features (i.e, genes, clinical markers, etc.) collected longitudinally from multiple patients (3 or more time points per patient). Importantly, patient time points can differ in number and

in collection time. Broadly, the TimeAx process consists of three steps (Fig. S1A, see *Methods*): First, a feature selection step, in which TimeAx uses either a user-predefined or an unsupervised, computationally-selected, set of features whose dynamics are loosely similar across patients (*"conserved-dynamics-seed"*). While not directly associated with the hidden disease progression axis, their shared dynamics point toward their ability to serve as the backbone for the comparison between patients' disease trajectories (Fig. 1B). Second, TimeAx builds a model, which approximates the shared dynamics across all patients' disease trajectories (Fig. 1C). Finally, TimeAx can leverage the model to identify the disease state of a particular individual at a particular time point, referred to as 'disease pseudotime'. By capturing a shared representation of disease progression dynamics across patients, TimeAx-inferred disease pseudotime can simulate patient-specific disease states, discovering disease progression-related mechanisms and providing predictive clinical utility. By explicit modeling of sample identity and sequential sampling as well as iterative building of the trajectory, TimeAx outperforms standard models using chronological time or trajectory inference methods (Fig. 1D; additional information in Supplementary Note 3).

## Disease pseudotime captures disease progression dynamics better than chronological time

To showcase the molecular and clinical benefits of explicitly modeling disease progression from longitudinal data, we contrasted disease pseudotime with chronological time as a measure of disease progression, in both simulations (see Supplementary Note 3 *and* Fig. S1C–G) and real-life human disease datasets, using various data types as input. We first focused on influenza infection, an acute disease which affects the immune-system over short periods of time. In this study, 17 healthy adults were challenged with influenza and then profiled longitudinally for whole blood gene expression by RNA-seq at 13–15 fixed time points within the first 108 h following infection (Fig. 2A, see *Methods*; total of 268 samples[7], denoted as 'Longitudinal influenza cohort'). As samples were collected at fixed times, similarly across all patients, chronological time points could not be used to differentiate between symptomatic and asymptomatic patients (Fig. S2A). On the other hand, based on TimeAx modeling, we projected disease pseudotime positions for all samples, observing a clear gradual increase in disease pseudotime in symptomatic but not in asymptomatic patients (Fig. 2B and Fig. S2B; $p < 10^{-29}$), an observation which we validated using two additional cohorts, including both children and adult patients (Figure S2C–D; $p < 10^{-47}$ and 0.0003 for longitudinal adult and children cohorts respectively, see *Methods*). Moreover, we identified changes in molecular processes over the course of disease pseudotime that would otherwise be missed using chronological time as a basis for disease progression (Fig. 2C and Fig. S2E, see *Methods*). In addition to genes which were associated with both disease pseudotime and chronological time, we identified 3432 genes which were only significantly associated with disease pseudotime, accounting for ~29% of the genes and 79% of genes with any association. Of note, genes with positive associations to disease pseudotime were highly enriched for multiple pathways, including the interferon pathway and heme metabolism (Fig. S2F), which is in concordance with previous findings emphasizing these pathways as related to augmented immune responses[8–11] with implications to disease severity and patients' clinical outcomes during influenza infections.

Diseases often advance slowly and progress at different rates across patients, making it difficult to track their dynamics and understand their molecular drivers. A common approach is to cluster patient samples into disease subtypes, clinical or data-driven, which risks losing the continuous aspect of disease dynamics due to large differences in rates of disease progression between patients[12,13]. To highlight the ability of TimeAx to capture long-term processes of disease

progression through explicit quantitative modeling of disease dynamics, we studied urothelial bladder cancer (UBC). UBC is a tumor with high recurrence rates after cancer removal or treatment that predominantly presents as a non-muscle invasive tumor with a small proportion of patients progressing to its muscle-invasive form, increasing the risk of developing metastases[14]. We trained a TimeAx model using time series microarray data from 18 patients with recurring non-muscle invasive bladder cancer who ultimately progressed to advanced disease and who were sampled longitudinally during each incidence of tumor recurrence (Fig. 2D, see *Methods*; 'UBC longitudinal cohort'). In this cohort, each patient had 4–6 samples, collected up to 15 years apart from first to last recurrence[12].

Using TimeAx, we inferred disease pseudotime positions for this cohort, which exhibited high patient-specific variability when considering the chronological time that had passed since their primary diagnosis (Fig. 2E). UBC disease pseudotime also uncovered strong molecular associations, which could not be observed when modeling the data using chronological time (Fig. 2F; using linear regression). Specifically, we identified 7484 genes (~32% of the genes and 95% of genes with detected signal) as significantly associated solely with disease pseudotime and not with the chronological time (Fig. 2F, *upper left quarter*). These included known clinical biomarkers of UBC progression such as *CCL2* and *IFITM2*, as well as negatively associated *SGPL1*, a marker linked to positive outcomes in cancer[15–17] (Fig. S3A). This signal enhancement was also observed at the pathway level, where a TimeAx based analysis identified stronger associations for known cancer-related processes such as the epithelial-mesenchymal transition (EMT)[18,19], TNFα signaling[20], interferon gamma[21] and G2/M cell cycle checkpoint[22] (Fig. S3B, see *Methods*; $q < 10^{-48}$, $q < 10^{-25}$, $q < 10^{-19}$ and $q < 10^{-9}$, respectively).

Dynamic modeling by TimeAx is applicable to multiple data types, including imaging technologies commonly used for patient diagnosis and monitoring in the clinic. To further demonstrate the scope of TimeAx's utility, we sought to apply TimeAx to age-related macular degeneration (AMD), a chronic disease monitored periodically using low cost and non-invasive optical coherence tomography (OCT)[23]. AMD is an irreversible progressive chronic disease of the retina, resulting in decreased visual acuity and is one of the leading causes of blindness in developed countries[24,25]. We generated a TimeAx model of AMD progression, using segmented features, generated from OCT scans of 157 patients, each with 15 to 79 consecutive scans over several years (4953 scans overall; AMD train cohort). We then used the generated model to predict disease pseudotime positions for an additional 34,836 OCT scans, collected from 1641 different patients (Fig. 2G, see *Methods*; 2–90 consecutive scans per patient; denoted as 'AMD test cohort'[26]). The original analysis, based on OCT scans from a subset of both cohorts, used chronological time but did not identify any changes in retinal morphology associated with disease progression[26]. Consistent with this finding, we observed that individuals' disease pseudotime generally increased over time, and was highly variable between patients (Fig. 2H), suggesting that disease pseudotime accounts for patient variability—which is only partly captured by chronological time. Indeed, we observed a strong association between the increase in disease pseudotime and the severity of the patients' disease burden as assessed by visual acuity (Fig. 2I, top), while no significant association was found for chronological time (Fig. 2I, *bottom*). Moreover, we observed an increase, at higher disease pseudotime positions, in the usage of anti-VEGF injections —a clinical procedure for reducing the accumulation of retinal fluids that are associated with worst visual acuity (Fig. S4A; $p < 10^{-192}$). This supports the notion that retinal fluids appear, mostly, in late stages of the disease (see Supplementary Note 4 for full the AMD progression analysis, including the identification of disease progression-related segmented features and clinical applications). Taken together, these results suggest that

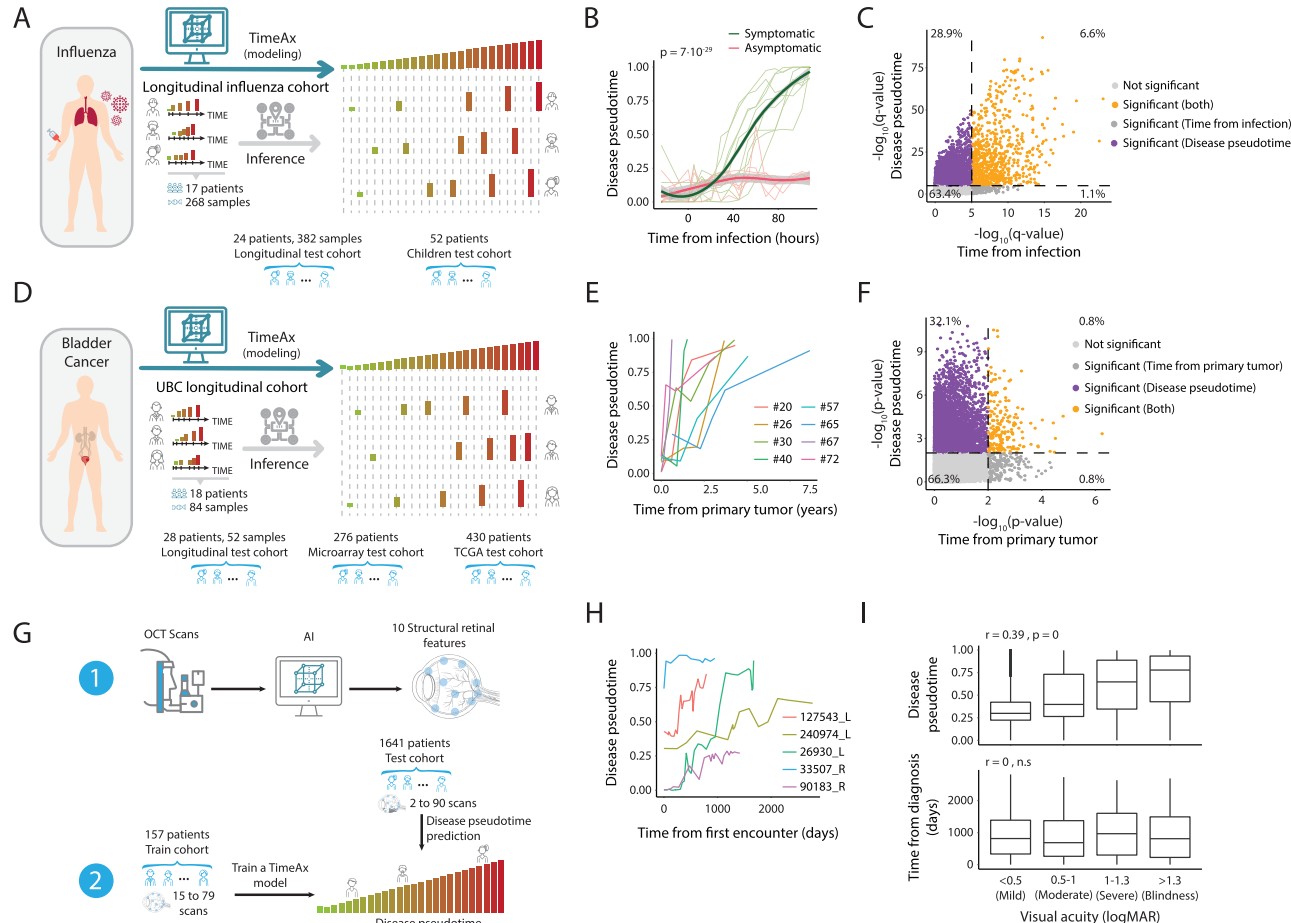

**Fig. 2 | Disease pseudotime captures disease progression dynamics better than chronological time. A** An illustration of the influenza infection dynamics TimeAx modeling and disease pseudotime inference, based on the longitudinal influenza cohort. **B** TimaAx disease pseudotime (y-axis) is different from chronological time (time from infection, x-axis). Shown are the differences between symptomatic (green lines) and asymptomatic (red lines) patients from the longitudinal influenza cohort. P value was calculated by comparing the prediction of disease pseudotime (by ANOVA), using only time or the interaction between time and symptoms as predictors. Trend lines represent the average levels in each of the groups ± standard error. **C** Gene associations (−log10 transformed, FDR-corrected, Q values based on linear regression), across all symptomatic and asymptomatic patients, using either sampling time (x-axis) or disease pseudotime (y-axis), and applying a Q value threshold of $10^{-5}$ (Dashed lines) (See *Methods*). Genes are colored based on their association with the two time axes. **D** An illustration of the UBC dynamics TimeAx modeling and disease pseudotime inference, based on the UBC longitudinal cohort. **E** TimeAx disease pseudotime (y-axis) is different from chronological time (time from primary tumor, x-axis), exemplified in six patients in the

UBC longitudinal cohort. **F** Gene associations (−log10 transformed, P values based on linear regression) with sampling time (x-axis) and disease pseudotime (y-axis), using a P value threshold of $10^{-2}$ (Dashed lines) (See *Methods*). Genes are colored based on their association with the two time axes, displaying significant associations almost entirely only with the disease pseudotime. **G** An illustration of the AMD dynamics TimeAx modeling and disease pseudotime inference, based on segmented features extracted from OCT scans of the patients' retina. Optical coherence tomography (OCT). **H** TimeAx disease pseudotime (y-axis) is different from sampling time (time from first encounter, x-axis), exemplified in five patients in the AMD train cohort. **I** The distribution of AMD test cohort patients' disease pseudotime positions (top; n = 29205 biologically independent samples) and times from diagnosis (bottom; n = 11075 biologically independent samples) (y-axis), across different visual severity states, determined according to the patients' visual acuity levels (logMAR; x-axis). Boxes represent the 25th, 50th, and 75th percentiles; whiskers show maxima and minima. P values were calculated based on linear regression. We thank Yuval Abraham for his contribution in the design and creation of (**A**, **D** and **G**).

TimeAx dynamic modeling enables molecular interpretability and may provide increased clinical utility compared to naive longitudinal monitoring based solely on chronological time.

## TimeAx uncovers an advanced tumor state with unfavorable clinical outcomes

To further highlight the utility of TimeAx, we focused on the UBC progression model where the long time scales of disease progression result in large differences between patients' disease progression rates on the one hand, and on the other, the high resolution molecular data enables the study of cellular and molecular mechanisms of tumorigenesis. To confirm that our TimeAx model captures tumor progression, we compared the disease pseudotime positions in the UBC longitudinal cohort with the tumor stage of the patients (see *Methods*;

based on TMN staging), and saw a clear association between increased disease pseudotime and more advanced stages (Fig. 3A, p < 0.0005 by linear regression), compared to no association using chronological time (Fig. S5A). We validated these results using held-out longitudinal UBC samples from the same dataset, and two additional cross-sectional UBC test cohorts – the former consisting of microarray data of 276 UBC patients[27] and the latter of 430 UBC RNA-seq samples from the Cancer Genome Atlas (TCGA) Program (Fig. S5B, see *Methods*; respectively denoted as 'longitudinal', 'microarray' and 'TCGA' test cohorts). Tumors with higher disease pseudotime positions were also more transitional/invasive (Fig. S5C; $p < 10^{-7}$), compared to more papillary tumors found at lower pseudotime positions. In addition, patients with prior malignancy were positioned further along disease pseudotime (defined by TCGA; Fig. S5D; $p < 10^{-4}$).

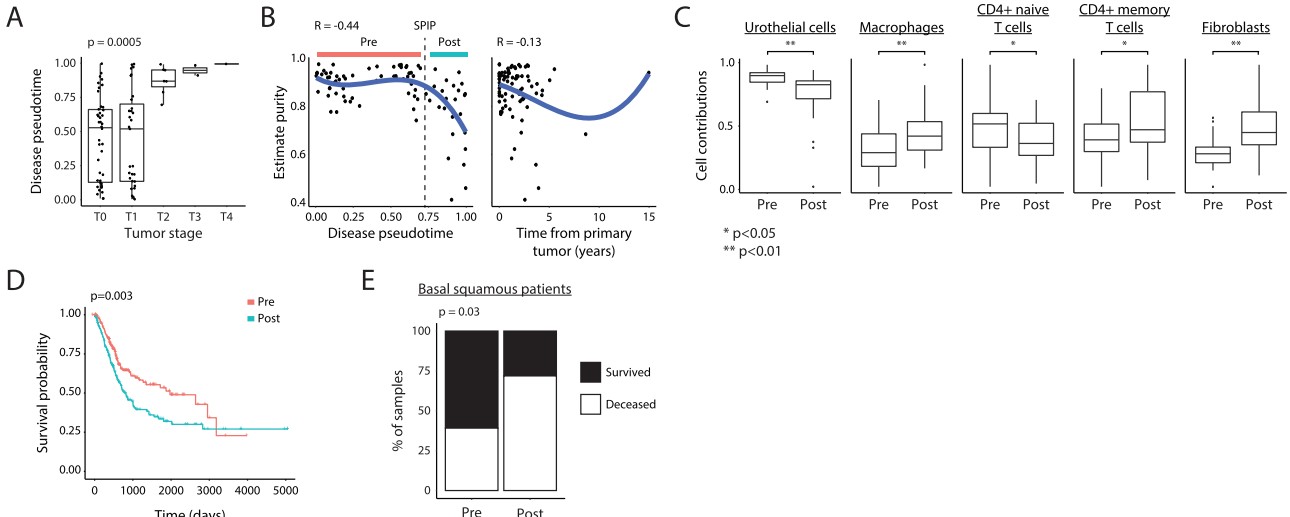

**Fig. 3 | TimeAx uncovers an advanced tumor state with unfavorable clinical outcomes. A** Disease pseudotime clinical applicability. Shown are disease pseudotime (y-axis) relation with tumor stage (x-axis) for samples within the UBC longitudinal cohort. P value was calculated based on linear regression. **B** Tumor purity scores (y-axis) along the disease pseudotime (left) and the time from primary tumor (right) (x-axis), displaying a sharp decrease in high disease pseudotime positions (SPIP; dashed line). Disease pseudotime ranges pre and post the SPIP are marked by colored bars. **C** Cell type deconvolved cell contributions (y-axis), displaying major differences between pre- and post- SPIP samples. * $p < 0.05$, ** $p < 0.01$ based on a two-sided t-test ($p < 10^{-4}$, 0.003, 0.05, 0.05, $10^{-4}$; Urothelial cells,

Macrophages, naive T cells, memory T cells and Fibroblasts, respectively). **D** Survival plot for UBC patients within the TCGA test cohort, comparing patients' tumors with disease pseudotime positions lower and higher the SPIP (pre versus post, respectively; color coded). P value was calculated based on a log-rank test. **E** Survived (black) versus deceased (white) percentages of basal/squamous patients within the TCGA test cohort pre and post SPIP (x-axis). p value was calculated using Fisher's exact test. In (**A** and **C**), boxes represent the 25th, 50th, and 75th percentiles; whiskers show maxima and minima and n = 84 biologically independent samples. Stromal pro-invasion point (SPIP).

We hypothesized that the TimeAx model, based on whole bulk-tissue transcriptomes, represents a continuous shift in complex multicellular programs in which both cell compositions and cell-states may change along the disease pseudotime and result in tumor progression. To explore our hypothesis, we first looked at tumor purity, a measure for the fraction of cancer cells in a tumor sample (calculated as in ref. 12), along the TimeAx UBC disease progression model. We identified a position along the disease pseudotime (disease pseudotime of 0.7), where a sharp decrease in tumor purity occurred, a trend undetectable when ordering samples by the time from primary tumor occurrence (Fig. 3B, see *Methods*; r = −0.44 and r = −0.1, respectively). A decrease in tumor purity has been previously associated with elevated immune infiltration and overall poor prognosis[28,29]. This sharp decrease in tumor purity was not associated with patients' demographics, including sex (Fig. S5E) and age (Fig. S5F). Taken together, the TimeAx model highlights the existence of a stromal pro-invasion point (SPIP), characterized by a change in the immune-stroma composition within the tumor microenvironment[29,30].

To understand how cellular composition and regulatory programs change along the disease pseudotime and how these relate to the decrease in tumor purity at the SPIP (Fig. 3B), we deconvolved the cell composition of all samples in all four cohorts and, predicted the compositions of seven major cell types: urothelial cells, muscle cells, basal tumor cells, endothelial cells and fibroblasts, as well as T cells, macrophages and additional immune cell subtypes (see *Methods*). Samples localized to disease pseudotime positions 'beyond the SPIP' showed a significant decrease in the abundance of urothelial cells, accompanied by an increase in the abundance of activated macrophages and fibroblasts (Fig. 3C and Fig. S5G; $p < 0.003$), a transition from naive to memory CD4 + T-cells (Fig. 3C and Fig. S5G; $p < 0.05$) and no significant change in basal tumor cells (Fig. S5G–H; $p > 0.1$). We reasoned that differences in cell compositions along the TimeAx model may be due to either differences in disease severity or technical variation during biopsy sampling. We therefore decided to leverage the clinical outcome data available in the TCGA test cohort to test

whether mapping pre- versus post- the SPIP carried clinically meaningful signal (see *Methods*). Indeed, we observed a significantly lower survival probability for patients mapping to the TimeAx disease progression axis past the SPIP (Fig. 3D; $p < 0.003$ by Kaplan-Meyer). The classification of patients into either pre- or post- SPIP improved survival rate prediction even after accounting for other covariates with known association with survival, including age, sex and the clinical stage of the disease (Supplementary Data 1; $p < 0.02$, Cox Proportional-Hazards Model). Specifically, even within a specific molecular cancer subtype, such as urothelial-like and basal/squamous tumors, patients with disease pseudotime positions past the SPIP displayed a higher mortality percentages (Fig. 3E; 72% compared to 39%, $p < 0.03$) and more rapid rates of mortality (Figure S5I, J). These observations suggest that the SPIP we observed reflects a biological milestone in UBC progression, in line with previous findings associating the infiltration of activated immune cells and cancer-associated fibroblasts into the tumor microenvironment with tumor progression and poor clinical outcome[31,32]. Taken together, this suggests that the TimeAx model accurately reflects tumor developmental processes and improves clinical prediction over the previously suggested tumor classification frameworks.

### Disease pseudotime captures variation undetectable by current stratification frameworks

Current clinical assessment of UBC progression, as well as the selection of interventions and therapies, relies mostly on histopathological staging. As patients' disease courses in UBC are highly heterogeneous, these traditional methodologies, focusing on a relatively small set of markers, are not sufficient for optimal clinical decision making. Recently, molecular profiling analyses divided urothelial carcinomas into two major molecular types, luminal and basal, with the latter showing down-regulation of urothelial differentiation markers, a higher incidence in muscle invasive tumors and association with worse clinical prognosis[33]. The luminal type, which represents most of the non-muscle invasive tumors, can be further divided into urothelial-like

and genomic-unstable subtypes, with the former harboring alterations in the FGFR3 pathway[34–36]. While these subtyping frameworks aim to stratify patients, they are not widely applied in the clinic due to their high complexity and uncertainty surrounding their utility for clinical prognosis over traditional frameworks[33]. Importantly, the grouping of patients into small sets of disease subtypes results in the loss of continuity in assessing disease progression. For example, though the original analysis[12] of the 'UBC longitudinal cohort' (Fig. 2D and *Methods*) divided patients' UBC recurrences into distinct molecular subtypes, it also showed that clinical progression occurred in all patients regardless whereas molecular subtyping remained predominantly stable in most patients. Moreover, in some cases patients were simultaneously associated with two different subtypes[12]. This suggests that discretized molecular subtyping does not reflect disease state, nor necessarily severity, and in some cases, dynamical changes in disease progression may be interpreted as different disease subtypes.

We hypothesized that the disease pseudotime we identified represents a generalized UBC disease dynamics axis, shared across luminal and basal molecular subtypes. Indeed, relying on previously published molecular typing ('LundTax' tumor molecular classification[37]), modeling disease dynamics, while excluding patients with basal tumors, allocated all patients to nearly the same disease pseudotime positions (Fig. S6A; $r = 0.91$), and yielded similar molecular associations as in the original model (Fig. S6B). This suggests that these molecular subtypes reside in the same disease dynamics axis, which accommodates transitions between molecular subtypes along the disease pseudotime. Supporting this, we observed higher disease pseudotime levels in more progressive molecular tumor subtypes, such as basal and mesenchymal, compared to lower values in urothelial-like subtypes (Fig. 4A; $p < 10^{-5}$ by one-way anova). This classification by molecular subtypes was strongly associated with patients' allocation to either the pre- and post- SPIP groups (Fig. 4B) but the resolution achieved by this classification was insufficient for the correct assignment to patients' histological staging (Fig. S6C, D).

We reasoned that in the absence of long-term molecular follow up of patients, understanding the relationship between disease progression and disease subtypes can only be garnered through further mechanistic understanding and the observation of clinically actionable predictions. As early events in UBC tumorigenesis are largely unresolved, we focused on disease pseudotime positions prior to the SPIP. TimeAx allowed us to discern high resolution dynamics and observe differences between early (primary) and recurring tumors with a large variation in disease pseudotime between recurrent tumors (Fig. 4C; $p < 10^{-9}$). Considering the present UBC molecular subtypings, we observed that while most samples pre-SPIP were classified as low grade tumor stages (Lum-P, Lum-U by the consensus molecular subtyping[13] and Urothelial-like and genomically-unstable by Lund taxonomy[37]), those patients showed large variation in disease pseudotime (Fig. 4D and Fig. S6E). These observations suggest that the current molecular subtyping systems of UBC tumors only allow for a coarse stratification of patients, precluding the high resolution granularity provided by modeling the continuous dynamics of disease progression.

Of all molecular subtypes, UroA exhibited the largest variation along the disease progression axis whereby UroA tumors at higher disease pseudotime positions were associated with lower survival percentages (Fig. 4E; $p < 0.01$). Consistent with this, by dividing the UroA tumors pseudotime continuum to 'early' and 'late', respectively, using disease pseudotime cutoff of 0.25, we observed lower survival percentages in late tumors (Fig. S6F; $p < 0.1$)–suggesting that these tumors represent different UBC progression stages.

## Disease pseudotime uncovers molecular mechanisms promoting UBC progression

We next explored whether we can detect molecular patterns associated with differences between early and late progression within the

UroA tumors, and whether those differences manifest different oncogenic transformation stages. We identified 2642 differentially expressed genes between early and late UroA progression tumors, which were highly co-regulated into two main modules ($q < 0.05$, Fig. 5A and Supplementary Data 2), one downregulated and one upregulated along the disease progression axis. In contrast, we observed no significant changes, between the two groups of UroA tumors, in the expression of established urothelial markers, including *CCND1*, *FGFR3*, *FOXA1*, *RB1*, *CDKN2A*, *GATA3*, *ERBB2*, *PPARG* and *XBP1* (Fig. S6G).

The downregulated module was composed of 1587 downregulated genes within late UroA tumors (Fig. S6H). This module was highly enriched in pseudogenes and microRNAs, ($q < 10^{-4}$, 0.03 respectively; Fig. 5B, see Supplementary Data 3 for functional enrichment). Interestingly, we noted that for many of these pseudogenes, their coding paralogs were transcribed by RNA polymerase III[38] and involved in biosynthetic processes promoting cancer cell proliferation[39], suggesting pseudogenes actively suppress the transformation process by downregulation of their coding equivalents. In addition, we noted an enrichment for protein coding genes in this module transcribing membrane channel proteins that were downregulated along disease pseudotime. Specifically, these included ligand gated ion channels ($q < 0.05$, Fig. 5C, see Supplementary Data 4), primarily calcium, potassium and sodium voltage- gated channels, controlling cellular ion homeostasis and downstream cell cycle and cell death processes. Interestingly, we also detected down regulation of G protein-coupled receptors (GPCRs) including neurotransmitter, hormone and free fatty acids receptors ($q < 0.05$, Fig. 5C, see Supplementary Data 4). GPCR associations with tumor progression have remained unclear, with evidence showing both tumor progressor[40] and suppressor[41] functions, likely explained both by differences in functionality between GPCRs and tissue and malignancy dependency. Our analysis showing downregulation of GPCRs as the disease shifts toward advanced UroA, pinpoints these GPCRs as likely tumor suppressors in bladder cancer.

The upregulated disease progression module contained 1055 genes (Fig. S6I), and was highly enriched for pathways associated with malignant transformation as well as known hallmarks of cancer including sustained proliferative signaling, activating invasion and metastasis, genome instability and mutation and deregulation of cellular energetics ($q < 0.05$; Fig. 5D, see Supplementary Data 5). Specifically, we detected a striking upregulation of genes in the ubiquitin proteasome (UPS) system including structural components of the 26 S proteasome, components of the anaphase promoting complex and ubiquitin ligases. Post-translational polyubiquitylation of key regulatory proteins, results in their proteasomal degradation and alters regulation of cell cycle and epithelial to mesenchymal transition[42]. In addition, late UroA tumors showed upregulation of genes functioning in cellular metabolism, including autophagy and oxidative phosphorylation, allowing the tumor to meet the increasing energetic demands and support cell proliferation during oncogenic transformation[43].

Interestingly, we detected that the shift from early to late UroA tumor progression was associated with an abundance of genes involved in mitotic kinetochore- spindle microtubules (MT) interaction which suggested it as a mechanism for malignant transformation in UBC, undetectable without TimeAx modeling. These genes spanned six integral kinetochore constituents[44] which together suggest that the shift from early to late UroA involves increasing chromosomal instability and aneuploidy. Specifically, these include MIS12[45], Ska[46,47], RZZ[48] components of the outer kinetochore, components of the nuclear pore complex NUP107−160[49,50] localized to the kinetochore during mitosis, the MT binding protein CLASP-2 and the chromatin remodeler RSF1[51], all required for stabilizing dynamic MT to kinetochores[52]. Similarly, late UroA tumors showed upregulation in genes coding for proteins localized to the spindle, including a subunit

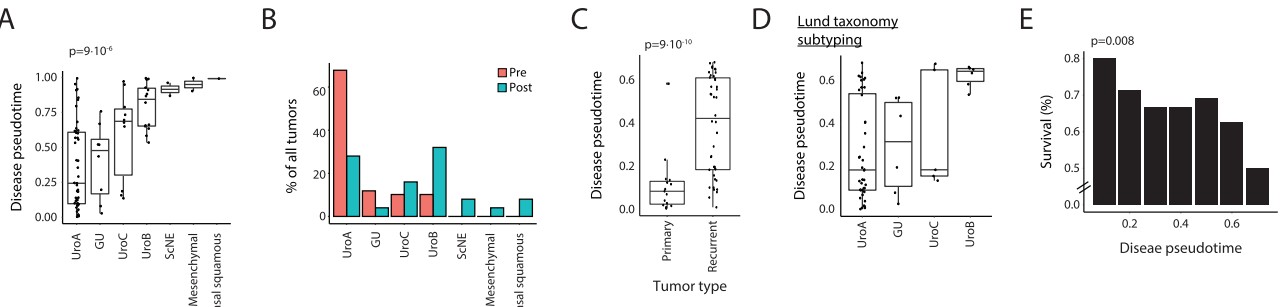

Fig. 4 | Disease pseudotime captures variation undetectable by current stratification frameworks. A Disease pseudotime distribution across tumor molecular classifications in patients within the UBC longitudinal cohort based on the 'Lund-Tax' molecular subtyping framework. B Distribution of 'LundTax' tumor molecular classifications for pre (red) and post (blue) stromal pro-invasion samples within the UBC longitudinal cohort. C Comparison of disease pseudotime (y-axis) between primary and recurrent tumors in pre-SPIP samples within the UBC longitudinal cohort. D Disease pseudotime distribution across tumor molecular classifications in patients pre-SPIP within the UBC longitudinal cohort based on the 'LundTax' molecular subtyping framework. E Percent of surviving patients in the TCGA test cohort with UroA tumors (y-axis) across disease pseudotime bins (bin size = 0.1; x-axis) pre-SPIP. p value was calculated based on linear regression. In (A, C and D), boxes represent the 25th, 50th, and 75th percentiles; whiskers show maxima and minima and n = 84 biologically independent samples.

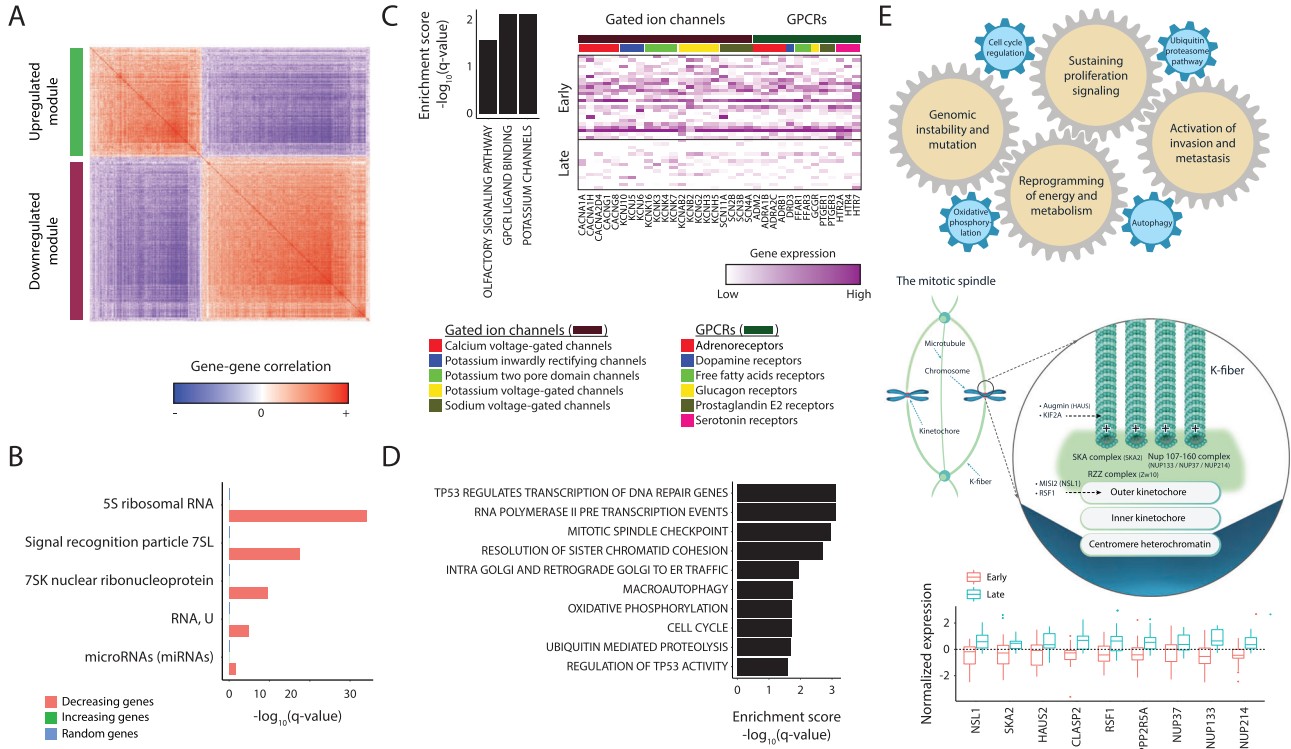

Fig. 5 | Disease pseudotime uncovers molecular mechanisms promoting UBC progression. A Co-expression matrix for genes (rows, columns) differentially expressed (q < 0.05) between early and late UroA tumors. Two gene sets were discovered and are highlighted in green and purple. B High enrichment of different pseudogene families within the downregulated module, compared to the upregulated module set and a random gene set (color-coded). C Pathway enrichment for the downregulated module, including pathway enrichment scores (left) and a heatmap of the expression levels of these pathways genes (columns), in early and late UroA tumors (rows) within the UBC longitudinal cohort. D Pathway enrichment scores for the upregulated module. E An illustration of the molecular model of UBC tumorigenesis, discovered by the TimeAx-based analysis, based on the increasing gene module. The illustration includes the association of known hallmarks of cancer with disease pseudotime (top) and the suggested mitotic kinetochore-spindle microtubules (MT) interaction (middle), which is also presented as box-plots of differential expression of its subunits (bottom; n = 40 biologically independent samples, boxes represent the 25th, 50th, and 75th percentiles; whiskers show maxima and minima). G protein-coupled receptors (GPCRs). We thank Yuval Abraham for his contribution in the design and creation of (E).

of the augmin complex (Haus) functioning in MT end capture[53] and KIF2A functioning in MT depolymerization, bi-polar spindle formation and chromosome pulling[54] (Fig. 5E and Supplementary Data 2). Taken together the signal detected via TimeAx disease progression modeling versus molecular subtyping suggests that modeling of disease progression not only provides high utility for understanding mechanistic changes in the biology of disease but is likely a necessary step prior to unsupervised sub-typing of diseases.

## Discussion

We present TimeAx, a framework for studying time-dependent disease dynamics at high resolution. TimeAx modeling frees researchers from relying on chronological time in experimental designs and analyses. Instead, it makes biological time (disease pseudotime) a comparable unit that researchers can discuss quantitatively. Akin to the insight introduced by sequence alignment[6], and single cell cell-state trajectory inference[55], TimeAx opens the door to a high-resolution understanding

of complex disease dynamics, often hidden due to the heterogeneity in patients' disease courses. Specifically, based on different omics technologies, including sparse data, TimeAx allows the inference of a disease pseudotime position—a quantitative metric that represents the samples' disease progression state—which can be used to build a high-resolution understanding of the temporal disease process, facilitating mechanistic discovery and patients' outcome prediction.

We highlight TimeAx's utility for both acute and chronic diseases, including the host responses after challenging healthy individuals with influenza virus, AMD disease progression and UBC tumorigenesis. In all cases, we discovered a high degree of molecular regulation, which could not be captured based on the analysis using chronological time and showcased the clinical utility of incorporating biological disease progression for patient diagnosis and disease prognosis. We further demonstrated TimeAx utility for modeling diverse data types, including not only omics data, such as transcriptomics from blood or biopsy, but also incorporating data used regularly in the clinic, such as segmented features extracted computationally from OCT scans of patients over time.

Even though the underlying mechanisms of disease progression are shared across many patients, its manifestation might vary in time and magnitude within each patient trajectory—affecting the utility of chronological time as a measurement of disease progression. On the other hand, disease pseudotime presents high variation over time, capturing hidden shared dynamics. Moreover, we note that one may use this framework to uncover patient-specific disease trajectories through repeated assembly of the trajectory with samples from a single patient left out. Combining these trajectories with chronological time should allow the calculation of disease progression rates, which comes with major implications for predicting patient prognosis and patient-specific treatment.

We followed the progression dynamics of UBC recurrences, across patients over many years. Even in the expected presence of many small-accumulated environmental effects over the long lifespan of the study, we were able to devise a continuous disease progression trajectory that captured the shared dynamics across all patients. The disease pseudotime, inferred by the model, supported patient stratification at a higher resolution compared to the patient stratification systems currently used in the clinic while demonstrating improved patient outcome prediction. In addition, it captured cellular and molecular mechanisms of disease progression, which were hidden due to patients' disease course heterogeneity. Specifically, we observed a major drop in tumor purity at late pseudotime positions (denoted as the 'stromal pro-invasion point'; SPIP), which is associated with the remodeling of the tumor microenvironment including the transition from naive to activated immune cells and an increase in fibroblasts. We show that patient transitions between pre- to post- SPIP pseudotime positions result in increased disease severity, reflecting worse patient outcome, even when patients are classified among identical molecular subtypes. These observations emphasize the utility of the continuous disease progression model over previously suggested patient stratification frameworks.

While cell-compositional changes strongly affected the molecular profiles of the tumors, we were able to utilize the disease pseudotime to observe molecular processes which correspond to the deregulation of cellular programs associated with tumorigenesis, ultimately leading to increased tumor proliferation. Among the deregulated cellular programs, we observed loss of cell cycle regulation and metabolic reprogramming as shown by upregulation of genes from the ubiquitin proteasome system, autophagy and oxidative phosphorylation. Of particular interest, we observed an upregulation of a group of genes localized to the contact site between the mitotic kinetochore and the spindle microtubules (Fig. 5E). The upregulation of these genes could be the mere result of increased levels of mitotic cells, due to increased proliferation rates. Another possibility, however, is the unraveling of

an uncharacterized mechanism contributing to chromosomal instability (CIN) in bladder cancer. Merotelic kinetochore attachment is an erroneous process in which a single kinetochore is attached to microtubules originating from both spindle poles during mitosis. This process is considered to be one of the major drivers of aneuploidy in mitotic cells and therefore to chromosomal instability[56]. Correction of merotelic kinetochore attachment requires an accurately regulated rate of MT-kinetochores attachment (formation and resolution). Changes in this rate, by experimentally increasing the stability of kinetochore microtubule attachments, results in increased levels of lagging chromosomes in anaphase indicating that slight changes in stability during mitosis are sufficient to increase chromosomal instability[57]. UBC progression was associated with the upregulation of several genes localized to and potentially stabilizing the kinetochore-MT interaction, therefore increasing the rate of merotelic kinetochore attachment[44]. Furthermore, the correction of erroneous kinetochore-MT interactions occurs by Aurora B kinase phosphorylating KMN network components to reduce their affinity to MT. Our data points to the upregulation of *PPP2R5A*, which encodes the B56 regulatory subunit of the Serine/threonine protein phosphatase 2A in UroA late tumors. This phosphatase localizes to kinetochores and stabilizes the kinetochore-MT interaction, counteracting the activity of Aurora B kinase[58]. Clearly, extensive experimental validation is required to confirm the proposed mechanism for chromosomal instability in bladder cancer. However, this sole example demonstrates the great potential of utilizing TimeAx-based disease pseudotime to provide highly predictive models for a better understanding of the molecular mechanisms leading to malignant transformation and for the discovery of potential novel drug targets.

In the present study we have demonstrated TimeAx's utility for modeling dynamics as a one dimensional consensus trajectory, however, this does not exhaust the full potential of our framework. TimeAx can be further extended to deal with diseases displaying more complex dynamics (such as branching trajectories) as well as sub-populations in the data. Specifically, this can be done by adding pre-processing and sub-group identification stages (in a supervised or unsupervised manner), and then running TimeAx separately on each subgroup. Alternatively, in the case one of the groups is too small, this can be done by building the trajectory on one group and assessing how this trajectory is disrupted when new individuals from another disease subtypes are sequentially added[59]. The seed detection step, currently limited to the discovery of increasing or decreasing features, can be also extended to features exhibiting more complex dynamics over time. Another extension of TimeAx would allow the inference of disease dynamics based on data from less than three time points per individual or through meta-analyses of multiple datasets. TimeAx can also be used to model other types of dynamics, such as disease recovery over time and non-disease biological processes, such as immune age[60]. Last, while used here for gene expression and features extracted from bioimages, TimeAx can be applied to other data modalities, such as protein, microbiome and epigenetic data, as well as clinical data, including clinical markers regularly used in the clinic and multi-omic data from the same patients.

## Methods

### The TimeAx algorithm
TimeAx aims to model the entirety of a disease's dynamics and construct a quantitative framework through which one can better compare individuals' sample states as part of a dynamic process shared by all individuals under examination. TimeAx takes as input a compendium of measurement profiles with each profile describing a time point (required at least 3), sampled from a specific individual and assayed when the individual was undergoing a biological condition whose trajectory we are interested in delineating. Profiles are a quantitative snapshot of the abundance of different measured data types

(any kind of omics; e.g., genes or proteins). The snapshots from each individual are then aligned with snapshots from other individuals to construct a consensus trajectory that describes the dynamics of the biological process above.

The TimeAx algorithm can be divided into three steps:

1. *Conserved-dynamics-seed* selection: Identifying disease dynamics requires a common ground. We thus start off by choosing a set of *"conserved-dynamics-seed" features (seed -features)*. These features can be predefined by the user or can be computationally-selected by focusing on features *whose* dynamics are similar across individuals.

2. Multiple trajectory alignment: Similar to principles stemming from the mature field of DNA sequence alignment, TimeAx performs a multiple trajectory alignment (MTA), merging all individuals' trajectories into a unified consensus trajectory. The alignment process relies solely on the above described *"conserved-dynamics-seed"*

3. Disease pseudotime estimation: TimeAx allows the prediction of disease pseudotime for new samples, based on the set of consensus trajectories, generated during the alignment process.

Additional information about TimeAx is described in Supplementary Note 1.

## Public datasets

For influenza dynamic modeling, we trained the TimeAx model using data from an influenza challenge study (H3N2/Wisconsin strain) in which 17 healthy volunteers, between 18 and 45 years of age, were infected and then profiled longitudinally over 13–15 timepoints (a subset of 0, 5, 12, 21, 29, 36, 45, 53, 60, 69, 77, 84, 93, 101 and 108 h after infection) for whole blood gene expression by RNA-seq (total of 268 samples). All patients were treated by oseltamivir orally on a daily basis along the study[7] (Longitudinal influenza cohort; accession number G SE30550). To ensure that we capture disease progression dynamics, we trained the model using only the symptomatic patients (9 patients) but inferred disease pseudotime for all patients. We then validated our results using an additional data from a challenge study of an influenza infection (H1N1 strain) in 24 human adults, aged 20–35 years old, with similar study settings as in the longitudinal influenza cohort[61] (total of 382 samples; Longitudinal test cohort; accession number G SE52428) and a blood microarray data from healthy and H1N1 infected children (*n* = 19, 33 respectively), all below the age of 17[62] (Children test cohort; accession number G SE42026).

For the UBC disease progression model, we trained TimeAx using a time series microarray data of microdissected tumors collected from 18 recurring non-muscle invasive bladder cancer (NMIBC) patients (3 males and 15 females), at different events of tumor recurrence (4–6 samples per patient, collected up to 15 years apart from first to last recurrence). At least one full induction course of intravesical Bacillus Calmette-Guerin (BCG), an immunotherapy for early-stage bladder cancer, was given for some of the patients[12] (UBC longitudinal cohort; accession number G SE128959). To accurately capture the variation between tumor samples, we used a conserved-dynamics-seed containing 100 genes. The results for the UBC longitudinal cohort were validated in additional 28 patients (22 males and 6 females; total of 52 samples), which were excluded from the UBC longitudinal cohort due to having less than four samples per patient (denoted as 'longitudinal test cohort') and two additional test cohorts: a microarray gene expression data from the bladders of 276 UBC patients, who underwent radical cystectomy, not receiving any treatment before bladder extraction[27] (microarray test cohort; accession number G SE83586) and RNA-seq data from 430 UBC patients (116 females and 314 males), aged 34–90 with a median of 69, from the Cancer Genome Atlas (TCGA) Program (TCGA test cohort; downloaded from the TCGA data portal).

## The age-related macular degeneration (AMD) cohort

The AMD patient cohort and clinical data of the patients were provided from the database of the Department of Ophthalmology, Ludwig-Maximilians-University, Munich, Germany. To identify patients suffering from AMD, the data warehouse was searched for all patients with the appropriate ICD-Code. Diagnosis of AMD was confirmed after proof of typical morphological features such as Drusen in fundoscopy and OCT scans and/or of choroidal neovascularization in initial Fluorescein angiography (in case of neovascular AMD). Overall, the dataset contains longitudinal data of Optical computed tomography scans of 1798 patients with AMD. The study was approved by the institutional review board of the Department of Ophthalmology, Ludwig-Maximilians-University, Munich, Germany and adhered to the tenets of the Declaration of Helsinki. Written informed consent was obtained from each participant AMD patient prior to the intervention and all testing outlined herein. Ethics committee of the medical faculty of the Ludwig-Maximilians-University of Munich gave ethical approval for this work regarding the research on AMD.

For AMD dynamic modeling, we trained TimeAx using conserved-dynamics-seed of 10 previously segmented features (without conserved-dynamics-seed selection), including retinal atrophy, fibrosis, retinal thickness, epiretinal membrane, neurosensory retina, subretinal hyperreflective material, retinal pigment epithelium (RPE), fibrovascular PED, drusen and choroid, derived from OCT scans of 157 patients, each with 15–79 consecutive scans over the years (4953 scans overall; AMD train cohort)[26]. The segmented features were obtained using a deep U-net based semantic segmentation ensemble algorithm previously described in ref. 26. Using this model, we predicted disease pseudotime for additional 34836 OCT scans collected from 1641 patients (2–90 consecutive scans per patient; AMD test cohort), using the same segmented features[26]. In this dataset, patients were either not treated or treated by eye injections of anti-VEGF agents. Additional features, including posterior hyaloid membrane and intraretinal and subretinal fluids, were excluded from the model due large technical variation and the direct effect of treatment.

## Cell type deconvolution and single cell analysis

We inferred cell type compositional abundance by applying Cibersort[63] on UBC bulk gene expression profiles, using two different sets of signature profiles. The first set contained cell type profiles obtained directly from the UBC tumors and their microenvironments. Specifically, based on single cell RNAseq data from a muscle-invasive urothelial bladder cancer patient[64], we calculated the mean expression profiles of seven major different cell types: Urothelial cells (88 cells), T cells (369 cells), Muscle cells (186 cells), Basal tumor cells (436 cells), Endothelial cells (362 cells), Macrophages (283 cells) and Fibroblasts (273 cells). This set allowed us to focus on changes in the abundance of non-immune as well as the main immune cell subsets. To study higher resolution immune cell subsets, we used the LM22 reference matrix[63], which contains gene expression profiles of different subtypes of macrophages, T, B, NK and dendritic cells. This reference matrix was shown as reliable for deconvolving tumor samples[65].

In addition, we focused on basal tumor cells within the single cell data and imputed the main two axes of variation, based PCA analysis using the top 1000 highly variable genes (HVGs). We linked these axes to tumor progression by presenting the association between the mean expression of HVGs and that of epithelial-mesenchymal transition (EMT) genes, observing similar spread across the two axes.

## Gene associations with disease pseudotime

For each cohort in this study, positive and negative gene associations with disease pseudotime were calculated as 'Pearson' correlations between gene expression levels and the disease pseudotime positions

across all the cohort's samples. To compare the number of disease pseudotime versus sampling time associated genes, we calculated the FDR-corrected $p$ value ($q$ value) based on a polynomial regression between gene expression and either of the time axes using a single $q$ value cutoff ($10^{-5}$ and 0.01 for Fig. 2C and Fig. 2D, respectively). In Fig. S2F, we present a generalized comparison across multiple $Q$ value cutoffs ($q < 10^{-5}$, $10^{-4}$, $10^{-3}$, $10^{-2}$, $10^{-1}$).

## Pathway enrichment

To explore the relations of biological functions with the progression of the disease, we tested candidate pathways from the following sources: GO, MSigDB Hallmark gene sets, Reactom, and KEGG. We correlated the expression levels of each gene with the inferred disease pseudo-time across samples (Pearson's r score). Next, for each pathway, we calculated a significance score for the difference in distributions between correlation coefficients of pathway member genes versus background genes that are not members of the pathway, using a Kolmogorov–Smirnov test (KS test). Finally, to obtain enrichment scores and allow their comparison across pathways, we applied a −log10 transformation.

## Reporting summary

Further information on research design is available in the Nature Portfolio Reporting Summary linked to this article.

## Data availability

Public datasets used in this paper can be found at National Center for Biotechnology Information Gene Expression Omnibus (GEO) under accession numbers GSE30550, GSE52428, and GSE42026 (influenza) and GSE128959, GSE83586 and the Cancer Genome Atlas (TCGA) (UBC). The dataset of the AMD cohort generated and analyzed during the current study are not publicly available due to data protection reasons of sensitive clinical data. Access to the data could be granted as part of scientific collaborations with the Department of Ophthalmology, Ludwig-Maximilians-University (Ben Asani, M.D., ben.asani@med.uni-muenchen.de). Data Sharing is restricted under data protection law and will be fully anonymized before sharing. A response to any requests initiated generally might take up to 1 week. A time frame for an agreement is dependent upon the parties' respective legal departments and possible contracting issues but can last up to 6 months.

## Code availability

TimeAx is publically available as an R package at Github: https://github.com/shenorrLabTRDF/TimeAx (https://doi.org/10.5281/zenodo.8188514[66]). Source code for reproducing the UBC analysis and figures is available at Github: https://github.com/shenorrLabTRDF/TimeAxPaperCode.

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

## Acknowledgements

We thank Y. Abraham for help with designing and creating figure illustrations and Martin Lukacisin, Rebecca Bendayan and Tim Cooper for their valuable feedback and discussion. This research was supported by the ISRAEL SCIENCE FOUNDATION (grant No. 1626/20), within the Israel Precision Medicine Partnership program. This work was supported in part by the German Research Foundation (DFG) to J.L.S. under Germany's Excellence Strategy (DFG—EXC2151—390873048); the HGF grant sparse2big, the EU H2020 projects SYSCID (Grant Agreement No. 733100) and ImmunoSep (Grant Agreement No. 847422). J.L.S. was further supported by the BMBF-funded excellence project Diet–Body–Brain (DietBB) (grant number 01EA1809A), iTREAT (FKZ: 01ZX1902A), and by NaFoUniMedCovid19 (FKZ: 01KX2021, project acronym "COVIM"). This study was funded in part by the European Union's Horizon 2020 Research and Innovation Program under the ERA-Net Cofund action no. 727565; the Joint Programming Initiative, A Healthy Diet for a Healthy Life (JPI-HDHL; project 529051018) and under the ERA-CVD non-cofunded action JTC2017 (Mechanisms of early atherosclerosis and/or plaque instability in Coronary Artery Disease)) awarded to J.L.S. The results here are in part based upon data generated by the TCGA Research Network: https://www.cancer.gov/tcga.

## Author contributions

S.S.O. conceived the idea, A.F. and S.S.O. developed the TimeAx method, N.M. led the biological interpretation, A.A. and F.J.T. contributed to method development, A.F. performed the analysis. H.S., B.A., J.B.S., S.G.P., and F.J.T. contributed to the image analysis. E.B. and K.R.B. contributed to the simulation analysis. J.L.S. helped with biological interpretation. A.F., N.M., and S.S.O. wrote the paper and all authors reviewed and revised it.

## Competing interests

S.S.O. holds equity and is a consultant of CytoReason. A.F., E.B., and K.R.B. are employees and hold equity in CytoReason. F.J.T. reports receiving consulting fees from ImmunAI and CytoReason and ownership interest in Dermagnostix. S.P. receives speaker and consultant honoraria from and has served on advisory boards for Abbott, Alcon, Geuder, Oculus, Schwind, STAAR, TearLab, Thieme Compliance, Ziemer, Zeiss and research funding from Abbott, Alcon, Hoya, Oculentis, Oculus, Schwind and Zeiss. The remaining authors declare no competing interests.
