## [Peer Review File · Nature Communications]

Reconstructing disease dynamics for mechanistic insights and clinical benefitREVIEWER COMMENTS

Reviewer #1 (Remarks to the Author):

In this work, Frishberg et al. developed TimeAx, a method which captures a representation of disease progression dynamics over time using high-dimensional time series data, such as patient transcriptomes or features extracted from medical imaging. They achieved this by inferring a trajectory of disease progression, called “disease pseudotime”. They demonstrate using TimeAx on multiple disease and datatypes, such as influenza infection, AMD disease progression and UBC tumorigenesis. This approach could help in discovering underlying molecular mechanisms of the disease, improve patient stratification and outcome prediction.

Overall, this is an interesting work, impressive in its technical details and demonstrating novelty and clinical relevance. However, we raise several aspects in which this manuscript needs to be improved before being published in Nature Communications. We recommend “Major revision”.

Specific comments for the authors:

- 1) In the introduction, the authors should discuss more elaborately relevant previous work. Are there other models for disease stratification? If so, the authors should present an example of comparing the performance of TimeAx with existing tools.
- 2) The methods section, as it is currently written, is not accessible to audiences that do not have a substantial background in computational biology. The authors should rewrite the methods section such that it would be easier to understand for a wider audience and perhaps move some of the more technical explanations to the supplementary.

Specific comments for the methods section:

- o In line 589-590, please give an example of values used for number of time points and number of iterations.
- o Lines 598-617, please explain in terms that can be understood by readers that do not necessarily have a background in computational biology how the alignment process works.
- o The pseudocode starting in line 619 is not clear.

3) The authors mention that a key motivation for their work is the difficulty of capturing the high heterogeneity between patients with existing methods of patient stratification (e.g. line 33, line 51, line 78, Fig. 1A). How is this addressed in TimeAx? In Fig. S1B (also described in Supp. Note 1) they demonstrate the contrary, that their model cannot deal with patient heterogeneity.

4) The authors emphasize throughout the manuscript that current methods for clinical staging or subtyping of diseases can be inaccurate in capturing disease progression (e.g. lines 80-81, Fig. 1A). However, in their results they compare disease pseudotime only with chronological time (Fig. 2B, 2E, 2H, 2I, 3B). Can they also show the association between disease pseudotime and clinical subtyping of disease?

5) In the methods section (Lines 580-595) the authors describe the process of feature selection. If the features are selected by testing Spearman correlation between pairs of patients, how can they select complex features (Fig. 1B), that are neither monotonously increasing nor decreasing?

6) In Fig. 2, only Fig. 2G illustrates the usage of an external test set (unlike Fig. 2A, 2D that only show one dataset). However, in the methods section and in lines 116-118 the authors explain that they used external test sets for the other diseases as well. Please clarify.

7) How were Fig. 2B and 2E derived, if not by using an external test set? Was this done by cross-validation on the training set? Please explain in the results section pertaining to these figures.

8) In the influenza model, the authors observe an increase in disease pseudotime in symptomatic but not in asymptomatic patients (Fig. 2B, S2C, lines 114-118). However, in the methods section they mention that they only trained on symptomatic individuals (lines 739-741). Hence, it is by design that the model would only be able to capture disease progression in symptomatic individuals. To make the point the authors are trying to say here, they should train the model on both types of individuals.

9) In the influenza model, when identifying genes that are associated with disease pseudotime or chronological time (Fig. 2C), do they use only symptomatic individuals or asymptomatic as well?

10) Gene associations with disease pseudotime are calculated by using Pearson correlation between gene expression levels and the disease pseudotime positions (lines 793-795). How would this analysis handle genes that are not changing monotonously along the disease pseudotime trajectory?

11) In lines 143-144, the authors mention that disease pseudotime positions exhibited high patient variation with respect to the chronological time elapsed from their primary tumor diagnosis. However, in Fig. 2E most patients except for one (#33, green) exhibit very similar trends. Perhaps the authors should select different patients to demonstrate this point.

12) While time from infection (influenza model) and time from primary tumor (UBC model) can be compared between patients, time from first encounter in the AMD model does not seem to be a clinically relevant metric. Thus, to demonstrate that disease pseudotime captures the progression in visual severity states better than chronological time (Fig. 2I), the chronological-time-based samples should first be aligned by disease stage.

13) The authors show an association between increasing disease pseudotime and advancing stages of UBC (lines 190-191, Fig. 3A). Is there also an association between chronological time and disease staging?

14) When deconvolving the UBC dataset with LM22 (Fig. 3B, lines 783-785 in the methods section), have the authors used the LM22 signature as-is, or included additional cell-types to capture the tumor cell fraction? If using LM22, please add a reference to justify using a signature derived from healthy tissue to deconvolve cancerous tissue.

15) A transition from naïve to memory CD4 T cells (line 220, Fig. 3C) is associated with increasing age. Have the authors checked for confounding factors separating the samples they are classified as “post” versus “pre”?

16) Regarding Fig. 4D:

- o The authors mention that Fig. 4D shows a large variation in disease pseudotime within molecular subtyping, and that this observation suggests that the current molecular subtyping could benefit from modeling the disease progression (lines 284-287). This deduction is unclear.

- o Fig. 4D only shows the pre-stromal invasion point samples, this is not explained in the main text.

- o The order of the tumor molecular classifications in Fig. 4A and 4D is not the same (GU before UroC in 4A). Please fix this so the figures would match.

17) To demonstrate the contribution of disease pseudotime to survival outcome prediction (Fig. 4E), survival rates of patients need to be controlled for known confounding factors such as age, sex, disease stage.

18) The authors derive a trajectory robustness score (lines 672-699) but almost do not mention it throughout the manuscript. What does a lower or higher robustness score signify? Does a lower robustness score imply that there is high heterogeneity in the data, or that there might be branched trajectories within the data? Please elaborate how this metric could be meaningful for researchers who would like to use this tool on their own data.

Reviewer #2 (Remarks to the Author):

Frishberg et al submit a manuscript entitled 'Reconstructing disease dynamics for mechanistic insights and clinical benefit'. The study intends to capture the dynamics of disease progression ultimately to assist in diagnostics and therapeutics via a new algorithm they developed called TimeAx, ultimately designed to flatten high-dimensional clinical data across a time series. The algorithm is based on defining a conserved dynamic 'seed' across a feature set and then applying that across different heterogeneous patient disease trajectories to find alignments across a disease progression over time. The approach is rather straight forward and understandable because it touches on standard methodologies like manual or automated feature selection/bootstrapping (e.g. pathways are less granular to uncover similar dynamics than gene expression), endotype analysis (patient clustering into like clinical and molecular phenotypes), sequence alignment/decision trees as tools to synch patient clinical and molecular signatures, and PCA/pseudotime trajectories. The authors test their algorithm across simulations as well as real world influenza, bladder cancer and serial retinopathy datasets. These tests were also cross-validated across several respective datasets and included sanity checks via cibersort datastreams, imaging, and biological (pathway) enrichment.

While I think of TimeAx as more of a compiled workflow than a blockbusting tool per se, I appreciate the depth of thought and design that have gone into its development and roll out, as well as in simplifying its approach and operations. The tests and validations presented in this manuscript are robust and cross-checked multiple times. TimeAx is indeed available publicly as an R package at Github for community testing and development. The group of investigators are very strong in this computational modeling field.

On the other hand, the often graphical style of the figures in this manuscript do belie TimeAx's complexities and complicated execution to the average reader. In looking over the github repository, it's clear that this isn't a one-button approach, rather a step-wise workflow, with one step dependent on

some pretty steep learning via the previous step. It's difficult to imagine novice bioinformaticians and/or basic research labs entering the field of omic technologies or analytics to be able to pick up and use TimeAx in clinical settings, rather it's more likely to be useful to advanced systems biology or computational biology groups. These concerns do dampen my enthusiasm somewhat for TimeAx's broad appeal in the absence of an off-the-shelf type of software wrapper. The proof will be in the pudding when TimeAx is applied to more datasets in the field, especially across other omic technologies, rather than the selection of mainly transcriptomic datasets with clear clinical phenotype data contained within this manuscript.

Reviewer 1

1. In the introduction, the authors should discuss more elaborately relevant previous work. Are there other models for disease stratification? If so, the authors should present an example of comparing the performance of TimeAx with existing tools.

Our response: We thank the reviewer for the comment. Originally, as we wanted to address a mixed audience of both computational and non-computational readers, we thought that the simulation should not be a major part of the story. Therefore, we created a simple simulation pipeline that demonstrated that TimeAx is superior over using chronological time as the axis of disease progression. However, we now realize that this was not sufficient due to two main reasons: 1. We did not compare TimeAx to trajectory inference methods, which might be considered as its main alternatives. 2. We only simulated linear expression patterns over true pseudotime, while many nonlinear associations exist in real life. In the revised manuscript we address these points by completely revising our simulation pipeline to include a comparison with diffusion maps, a common method used for trajectory inference, and to allow both linear and non linear trends. The revised simulation scheme and the new results can be found in new Supp. note 3 and new Figure S1B-F.

2. The methods section, as it is currently written, is not accessible to audiences that do not have a substantial background in computational biology. The authors should rewrite the methods section such that it would be easier to understand for a wider audience and perhaps move some of the more technical explanations to the supplementary. Specific comments for the methods section:

- o In line 589-590, please give an example of values used for number of time points and number of iterations.
- o Lines 598-617, please explain in terms that can be understood by readers that do not necessarily have a background in computational biology how the alignment process works.
- o The pseudocode starting in line 619 is not clear.

Our response: We thank the reviewer for his suggestion. From this comment, in addition to comments from the second reviewer, we now realized that the manuscript was not easy to read for non-computationals and it required a major revision to the methods section. In the revised manuscript, the methods section includes only the most important sections. We tried to keep these sections as clear as possible and remove any mathematical phrases. We now added the extended methodologies to the Supp. materials as Supp. Notes 1-3, describing the TimeAx method, the robustness and the simulation analyses. In addition, we followed the reviewer's specific mentions and revised them accordingly.

3. The authors mention that a key motivation for their work is the difficulty of capturing the high heterogeneity between patients with existing methods of patient stratification (e.g. line 33, line 51, line 78, Fig. 1A). How is this addressed in TimeAx? In Fig. S1B (also described in Supp. Note 1) they demonstrate the contrary, that their model cannot deal with patient heterogeneity.

Our response: We are sorry for the confusion regarding patients heterogeneity and how it is addressed by TimeAx. We now see that heterogeneity can be also regarded as the different disease subtypes across a population. In our mind heterogeneity often represents

differences in biological processes' dynamics between subjects. We show that ignoring such temporal differences results in significant loss of signal and an under-utilization of time series data. However, we now realize that this phrase can cause confusion. Therefore, in the revised manuscript we refined the text to be more clear in this regard.

Beyond this, with respect to patient subgroups, we believe this important issue can be addressed either in pre-processing stages (e.g. by identifying sub-groups in the data in a supervised or unsupervised manner, and then running TimeAx separately on each) or in the case one of the groups is too small, in downstream analysis, by building the trajectory on one group and assessing how this trajectory is altered when a new individuals are spiked in. For example, we have recently shown in Frishberg et al. Cell Reports Med, 2022, where we built a trajectory on COVID recovered patients, that by using 'spiked' in data from a few individuals who did not recover from infection successfully, we identify features critical for recovery. We have now added the following sentence to the discussion: *"TimeAx can be further extended to deal with diseases displaying more complex dynamics (such as branching trajectories). Specifically, this can be done by adding pre-processing and sub-group identification stages (in a supervised or unsupervised manner), and then running TimeAx separately on each subgroup. Alternatively, in the case one of the groups is too small, this can be done by building the trajectory on one group and assessing how this trajectory is altered when new individuals from another disease subtypes are sequentially added"*.

4. The authors emphasize throughout the manuscript that current methods for clinical staging or subtyping of diseases can be inaccurate in capturing disease progression (e.g. lines 80-81, Fig. 1A). However, in their results they compare disease pseudotime only with chronological time (Fig. 2B, 2E, 2H, 2I, 3B). Can they also show the association between disease pseudotime and clinical subtyping of disease?

Our response: We thank the reviewer for this suggestion. To be clear, in the previous version of the manuscript we not only compared pseudotime with chronological time but also to clinical measures currently used in the clinic (visual acuity score - Figure 2I, tumor staging - Figure 3A). However, to showcase it even further using patient clinical subtypes, we have now added a new analysis demonstrating the lack of usability of UBC molecular subtypes to determine tumor stages in both our longitudinal cohort and the TCGA test cohort (new Figure S6C-D).

5. In the methods section (Lines 580-595) the authors describe the process of feature selection. If the features are selected by testing Spearman correlation between pairs of patients, how can they select complex features (Fig. 1B), that are neither monotonously increasing nor decreasing?

Our response: We thank the reviewer for the comment. It is correct that our current seed selection implementation revolves around Spearman correlations and therefore is highly tuned for the discovery of monotonous trends. We do sometimes capture non-monotonic features as they tend to be less random over time compared to completely unrelated features. However, due to the confusion around this, in the revised manuscript we decided to remove the claim that our seed detection framework allows capturing those features. We

therefore revised Figure 1B and the methods section (Now in Supp. Note 1) and added the notion of capturing non-monotonic features as a future improvement for our framework.

6. In Fig. 2, only Fig. 2G illustrates the usage of an external test set (unlike Fig. 2A, 2D that only show one dataset). However, in the methods section and in lines 116-118 the authors explain that they used external test sets for the other diseases as well. Please clarify.

Our response: We apologize for the lack of clarity regarding the test cohort. These existed in the original manuscript but were not clearly labeled. We now revised Figures 2A and 2D to present the validation cohorts for each disease.

7. How were Fig. 2B and 2E derived, if not by using an external test set? Was this done by cross-validation on the training set? Please explain in the results section pertaining to these figures.

Our response: We realize that the definition of train and test data was not clear along the manuscript. In general, disease pseudotime was projected not only for samples in the test cohorts but also to the samples used to generate the model (train cohorts). Figures 2B and 2E demonstrate the change in pseudotime for the train cohorts. Specifically, in the UBC model, we could not use the test cohorts for this analysis, as the train cohort was the only longitudinal data we had. Due to small sample size, dividing this cohort into train and test resulted in a less robust model. For the influenza model, we now replaced the old Figure S2D with a new Figure S2C, showcasing the robustness of the model in another longitudinal cohort.

8. In the influenza model, the authors observe an increase in disease pseudotime in symptomatic but not in asymptomatic patients (Fig. 2B, S2C, lines 114-118). However, in the methods section they mention that they only trained on symptomatic individuals (lines 739-741). Hence, it is by design that the model would only be able to capture disease progression in symptomatic individuals. To make the point the authors are trying to say here, they should train the model on both types of individuals.

Our response: We are sorry for the lack of clarity regarding this matter. We initially trained the model using all patients but exhibited a major decrease in its power to discover interesting new molecular associations. While the asymptomatic patients share the disease axis with the symptomatic patients, it seems like their disease progression is limited, thus adding them to the model results in the addition of non-progression related noise. To ensure the robustness of the model in the case of asymptomatic patients, we validated our results using two non-related dataset (new Figure S2C and Figure S2D) showcasing similar trends and ruling out overfitting.

9. In the influenza model, when identifying genes that are associated with disease pseudotime or chronological time (Fig. 2C), do they use only symptomatic individuals or asymptomatic as well?

Our response: In this analysis we use all samples since now we have a common ground (pseudotime). We now clarify this point within the revised manuscript and the figure legend.

10. Gene associations with disease pseudotime are calculated by using Pearson correlation between gene expression levels and the disease pseudotime positions (lines 793-795). How would this analysis handle genes that are not changing monotonously along the disease pseudotime trajectory?

Our response: We thank the reviewer for this comment. We now see that we were not clear regarding gene associations and how they were calculated along the manuscript. There are two different types of analyses that require two different association approaches. Pearson correlation was only used to determine positive and negative associations, suggesting increasing and decreasing patterns along the disease pseudotime. We now emphasize this point more clearly in the methods section.

Concurrently, we used linear regression to determine and quantify associations in order to compare between disease pseudotime and chronological time (Figure 2C, 2F and S2F). We now realize that this approach is lacking in capturing nonlinear associations. In the revised manuscript, we changed the analysis to use polynomial regression instead (new Figure 2C, 2F and Figure S2E). In addition, we expanded the simulation analysis to check the effects of combinations of linear and nonlinear gene associations with true pseudotime (new Figure S1C-G).

11. In lines 143-144, the authors mention that disease pseudotime positions exhibited high patient variation with respect to the chronological time elapsed from their primary tumor diagnosis. However, in Fig. 2E most patients except for one (#33, green) exhibit very similar trends. Perhaps the authors should select different patients to demonstrate this point.

Our response: We thank the reviewer for raising this point. Since our main emphasis was the variation between patients, we initially wanted to demonstrate the difference between fast progressors and slow progressors (#33). However, we now see that this skews the plot, hiding all the variation we see within the main group of patients. Therefore, we now removed #33 and added several new patients with unique disease trajectories (new Figure 2E)

12. While time from infection (influenza model) and time from primary tumor (UBC model) can be compared between patients, time from first encounter in the AMD model does not seem to be a clinically relevant metric. Thus, to demonstrate that disease pseudotime captures the progression in visual severity states better than chronological time (Fig. 2I), the chronological-time-based samples should first be aligned by disease stage.

Our response: Thank you for the important comment. The reviewer rightfully noticed that the time from first encounter used initially is not a good representation of disease progression. To correct this, in the new Figure 2I, we now include in this plot only patients with which we have the time from diagnosis information. Similar to the UBC model, we see no association with this time axis as well.

13. The authors show an association between increasing disease pseudotime and advancing stages of UBC (lines 190-191, Fig. 3A). Is there also an association between chronological time and disease staging?

Our response: We thank the reviewer for this suggestion. We now added a new Figure S5A demonstrating a weak association (not significant) between chronological time and disease staging.

14. When deconvolving the UBC dataset with LM22 (Fig. 3B, lines 783-785 in the methods section), have the authors used the LM22 signature as-is, or included additional cell-types to capture the tumor cell fraction? If using LM22, please add a reference to justify using a signature derived from healthy tissue to deconvolve cancerous tissue.

Our response: We thank the reviewer for the suggestion. We did not use the LM22 reference to capture tumor cell fractions, but to study the immune cell subtypes compositions within the tumor microenvironment. To follow differences in tumor cell fractions across patients, we mostly relied on the tumor purity measure and its association with disease pseudotime, demonstrated in Figure 3B. Regarding the second question, LM22 was indeed used before in many tumor studies, including the original authors of the Cibersort algorithm (Gentles et al. Nature Med. 2015). We now added an appropriate sentence and a reference in the methods section.

15. Transition from naïve to memory CD4 T cells (line 220, Fig. 3C) is associated with increasing age. Have the authors checked for confounding factors separating the samples the are classified as “post” versus “pre”?

Our response: We thank the reviewer for the important remark. We indeed did not check this initially but we realize that patient demographics could have been a major factor associated with disease pseudotime. In the new manuscript we address this concern by adding the new Figure S5E and Figure S5F, demonstrating no significant association between patients' age and sex and their classification as “pre” vs. “post”. The analysis of age was done only for the TCGA data, as the age information was missing in the other cohorts. There is probably a weak association between age and the risk of turning into “post” stages. However, it is obscured due to patient heterogeneity.

16. Regarding Fig. 4D:

- o The authors mention that Fig. 4D shows a large variation in disease pseudotime within molecular subtyping, and that this observation suggests that the current molecular subtyping could benefit from modeling the disease progression (lines 284-287). This deduction is unclear.
- o Fig. 4D only shows the pre-stromal invasion point samples, this is not explained in the main text.
- o The order of the tumor molecular classifications in Fig. 4A and 4D is not the same (GU before UroC in 4A). Please fix this so the figures would match.

Our response: We are sorry for the lack of clarity. In the revised manuscript we modified this paragraph to emphasize these points and improve clarity. We also revised the new Figure 4A as suggested by the reviewer.

17. To demonstrate the contribution of disease pseudotime to survival outcome prediction (Fig. 4E), survival rates of patients need to be controlled for known confounding factors such as age, sex, disease stage.

Our response: We thank the reviewer for this comment. We now realized that the wording “survival rate” in this panel can be confusing. In this specific case, survival rate means the ratio of surviving patients from all patients with Uro-like tumors within each pseudotime bin along the disease pseudotime axis of the TCGA cohort. In the revised manuscript, we now rename this y-axis “Survival (%)”. In an additional analysis, we compute a Cox Proportional-Hazards Model (Supp. Table 1 and lines 228-231) where we controlled for known confounding factors such as age, sex. We did it for the whole pseudotime axis (looking at pre vs. post). In 4E we simply want to show that this effect can be also related to small changes within a specific disease subtype, pre- inflection point.

18. The authors derive a trajectory robustness score (lines 672-699) but almost do not mention it throughout the manuscript. What does a lower or higher robustness score signify? Does a lower robustness score imply that there is high heterogeneity in the data, or that there might be branched trajectories within the data? Please elaborate how this metric could be meaningful for researchers who would like to use this tool on their own data.

Our response: We thank the reviewer for this comment. Based on this comment and others, we now realize that the old manuscript was a bit too technical and the robustness score did not help in making it less so. While this score provides a valuable tool to test the validity of the model, we believe it had little to no impact on the storyline provided in the manuscript. Therefore, we removed any indication of the robustness score from the main text and kept it as a new Supp. Note 2.

Reviewer 2

1. The often graphical style of the figures in this manuscript do belie TimeAx’s complexities and complicated execution to the average reader. In looking over the github repository, it’s clear that this isn’t a one-button approach, rather a step-wise workflow, with one step dependent on some pretty steep learning via the previous step. It’s difficult to imagine novice bioinformaticians and/or basic research labs entering the field of omic technologies or analytics to be able to pick up and use TimeAx in clinical settings, rather it’s more likely to be useful to advanced systems biology or computational biology groups. These concerns do dampen my enthusiasm somewhat for TimeAx’s broad appeal in the absence of an off-the-shelf type of software wrapper.

Our response: We thank the reviewer for this important comment. We would like TimeAx to be applicable to a broad range of users. TimeAx is actually a one step action where you create your model using a single function *modelCreation* based on two relatively simple inputs (a feature matrix and a patient ID vector). Since this model can be used to estimate disease pseudotime in any new dataset, it has to have a different prediction function *predictByConsensus*, which only requires the model and the new feature matrix as inputs. However, based on this comment, realizing that the github repository does not reflect this, we now changed its README page to be more clear and user-friendly.

In addition, based on this comment and others, we now realize that the manuscript was not easy to read for non-computationals and it required a major revision to the methods section. In the revised manuscript, the methods section includes only the most important sections. We tried to keep these sections as clear as possible and remove any mathematical phrases.

We now added the extended methodologies to the Supp. materials as Supp. Notes 1-3, describing the TimeAx method, the robustness and the simulation analyses.

2. The proof will be in the pudding when TimeAx is applied to more datasets in the field, especially across other omic technologies, rather than the selection of mainly transcriptomic datasets with clear clinical phenotype data contained within this manuscript.

Our response: We thank the reviewer for this comment. TimeAx can be applied to different omics technologies as well as other measures (as shown in the AMD analysis). In the revised manuscript, this is now emphasized both in the discussions as well as in github repository's readme file. To emphasize this point even further, we also changed Figure 1C and Figure S1A to demonstrate that TimeAx's input can be any 'high dimensional' data type, even one that can be sparse.

REVIEWERS' COMMENTS

Reviewer #2 (Remarks to the Author):

The authors have addressed my concerns, and along with their response to similar comments from reviewer one, have worked diligently to improve the readability of the manuscript, ease of use of the tools on GitHub, and wider discussion/applicability of the method.

Reviewer #3 (Remarks to the Author):

I thoroughly enjoyed reading the revised version of the paper. The authors have taken several steps to improve their paper, including addressing the issues raised in the previous review. The paper is now more accessible to a wider audience and provides more comprehensive results and analyses. I believe that this paper will make a valuable contribution to the field of studying disease progression dynamics and will be of great interest to researchers in this area.

They have revised the introduction to include a more elaborate discussion of relevant previous work, including a comparison of TimeAx with trajectory inference methods. The authors have also made significant changes to the methods section, rewriting it to be more accessible to a wider audience and moving technical explanations to the supplementary materials. Regarding the specific comments, the authors have addressed the need for a more detailed discussion of disease heterogeneity and its treatment in TimeAx. The authors have provided additional analyses to demonstrate the lack of usability of certain molecular subtypes in determining disease stages, and they have added a new analysis to compare disease pseudotime with clinical subtyping of diseases. They have also clarified the selection of features and the association analysis with disease pseudotime, using both Spearman correlation and linear regression or polynomial regression.

The authors have revised the figures to provide clearer information and address issues such as the usage of external test sets and the need for alignment of chronological time-based samples. They have also made changes to better represent patient variation and disease progression in certain figures.

The authors have made several clarifications and improvements throughout the paper, including addressing confounding factors, explaining the use of LM22 signature, and discussing the robustness

score. They have also modified the survival rate plot to clarify its meaning and conducted additional analyses to control for confounding factors.

Overall, the authors have responded to the comments from the previous review thoughtfully and made appropriate revisions to address the concerns raised. Based on these revisions, I recommend accepting the paper.

Reviewer 2:

The authors have addressed my concerns, and along with their response to similar comments from reviewer one, have worked diligently to improve the readability of the manuscript, ease of use of the tools on GitHub, and wider discussion/applicability of the method.

Our response: We thank the reviewer for the nice words and happy that we were able to address all the reviewer's comments.

Reviewer 3:

I thoroughly enjoyed reading the revised version of the paper. The authors have taken several steps to improve their paper, including addressing the issues raised in the previous review. The paper is now more accessible to a wider audience and provides more comprehensive results and analyses. I believe that this paper will make a valuable contribution to the field of studying disease progression dynamics and will be of great interest to researchers in this area.

They have revised the introduction to include a more elaborate discussion of relevant previous work, including a comparison of TimeAx with trajectory inference methods. The authors have also made significant changes to the methods section, rewriting it to be more accessible to a wider audience and moving technical explanations to the supplementary materials. Regarding the specific comments, the authors have addressed the need for a more detailed discussion of disease heterogeneity and its treatment in TimeAx. The authors have provided additional analyses to demonstrate the lack of usability of certain molecular subtypes in determining disease stages, and they have added a new analysis to compare disease pseudotime with clinical subtyping of diseases. They have also clarified the selection of features and the association analysis with disease pseudotime, using both Spearman correlation and linear regression or polynomial regression.

The authors have revised the figures to provide clearer information and address issues such as the usage of external test sets and the need for alignment of chronological time-based samples. They have also made changes to better represent patient variation and disease progression in certain figures.

The authors have made several clarifications and improvements throughout the paper, including addressing confounding factors, explaining the use of LM22 signature, and discussing the robustness score. They have also modified the survival rate plot to clarify its meaning and conducted additional analyses to control for confounding factors.

Overall, the authors have responded to the comments from the previous review thoughtfully and made appropriate revisions to address the concerns raised. Based on these revisions, I recommend accepting the paper.

Our response: We thank the reviewer for all the major comments raised in the initial manuscript, pushing us for more accessibility and clarity. We are happy that we were able to address all the reviewer's comments and by that to overall improve the manuscript.